# Polarization-Angle-Insensitive Dual-Band Perfect Metamaterial Absorbers in the Visible Region: A Theoretical Study

Zhihui Xiong [1], Zhixi Li [1], Guangqiang He [2], Kecheng Su [3], Yien Huang [3] and Guowei Deng [4,*]

1 College of Physics and Engineering, Chengdu Normal University, Chengdu 611130, China
2 College of Electronic Engineering, Sichuan Vocational and Technical College, Suining 629000, China
3 Key Laboratory of Testing Technology for Manufacturing Process, School of Manufacturing Science and Engineering, Southwest University of Science and Technology, Mianyang 621010, China
4 College of Chemistry and Life Sciences, Chengdu Normal University, Chengdu 611130, China
* Correspondence: guoweideng86@163.com

**Abstract:** Metamaterial absorbers have been studied extensively due to their potential applications in the field of photonics. In this paper, we propose a simulation study of a polarization-angle-insensitive dual-band perfect metamaterial absorber with absorption peaks at 654 and 781 nm, respectively. By adjusting the structure parameters, dielectric thickness, and refractive index, the obtained absorber has high scalability in the visible wavelength region. To further understand the performance of the cross-structure absorber, analysis of its electric and magnetic field distribution shows that it produces two resonance modes leading to different absorption properties. In addition, the position and intensity of the absorption peaks were found to be unchanged with increasing incident polarization angle, indicating that the absorber is insensitive to the polarization of the incident light. The absorber has great flexibility and has good application potential in sensing and detection.

**Keywords:** plasmonics; metamaterial absorber; dual band; visible region

## 1. Introduction

In recent years, absorbers have garnered significant attention from researchers due to their crucial role in many applications, including sensors [1–4], thermal emitters [5], and imaging devices [6]. The design of dielectric absorbers has primarily relied on two types of microstructures. One approach involves utilizing Fabry–Perot-type thin-film structures, where a lossy dielectric film with a specific thickness is supported by a metal or concentrator to create a Fabry–Perot cavity. The arrangement allows for gradual absorption within the lossy dielectric layer [7–9]. However, the type of absorber has limited flexibility as the frequency and spacing of absorption bands are determined by the resonance conditions of the cavity and cannot be easily adjusted [10]. The other approach involves using metamaterial-based micro–nano structures [11–13], which entails surface plasmon resonances within metal nanostructures to achieve narrow absorption bands. Compared with thin-film absorber structures, the given metamaterial absorber structures have the advantages of miniaturization, high absorption efficiency, and tunability of absorption bands [14,15].

Metamaterials, as a new type of optical material, have gained widespread usage and rapid development in recent years due to their excellent optical manipulation capabilities [16–18]. They can be used as electromagnetic modulation devices, and metalenses in detection [19,20] and imaging applications [21–23], and they have controllable absorption properties from the visible light to microwave regions [24–27]. Many designs of metamaterial absorbers have been proposed in past studies. A commonly used design consists of a three-layer stack comprising metal–dielectric–metal layers that absorb incident light through plasmonic resonance achieved by adjusting the size and periodicity of metal nanostructures [28,29]. Most previously studied metamaterial absorbers only exhibit a

single resonant absorption band; there is a need for absorbers with multiple absorption bands for various applications such as biosensing [30,31]. In recent years, some plasma metamaterials with multiple absorption bands have been demonstrated [32,33]; however, most of the nanostructures studied are relatively small in size, monofunctional, unfavorable for processing, and costly; thus, metamaterial-based nanostructures are not fully utilized to their fullest potential.

In this paper, a polarization-angle-insensitive two-band perfect metamaterial absorber is proposed, which has strong absorption at 654 and 781 nm in the visible-light band, with absorption rates of 99.83% and 99.64%, respectively. By considering the impedance matching condition of the metamaterial absorber, it is possible to achieve near-perfect absorption by varying the size and dielectric thickness of the absorber. To realize the high absorption of dual frequency in the visible band, a cross-patterned structure with small dimensions was designed. The absorber has good geometrical tunability. The regular adjustment of visible-light absorption can be achieved by the design of the characteristic dimensions (L, W, h2) of the structure. The absorption mechanism at the resonant frequency is analyzed by calculating the electric and magnetic field distributions. In addition, the effects of refractive index and polarization angle on the absorption characteristics of the absorber are discussed. The proposed dual-frequency absorber has high absorption characteristics and is easily achieved using a simple structure.

## 2. Structure Design and Methods

The structure diagram of a polarization-angle-insensitive dual-band perfect metamaterial absorber is shown in Figure 1. The absorber consists of four layers, with the top layer composed of an array of Ag cross-cycles, followed by a $SiO_2$ dielectric layer, a Ag metal layer, and a Si substrate at the bottom. A Si substrate with a thickness of h4 = 200 nm and a relative dielectric constant of 3.4 is used to protect the device [34]. The transmission-blocking Ag reflector on the Si substrate has a thickness of h3 = 50 nm, resulting in near-zero transmittance [35]. The $SiO_2$ dielectric layer has a thickness of h2 = 100 nm. The dimensions for each unit within the Ag structure are as follows: length L = 130 nm, width W = 50 nm, and thickness h1 = 80 nm (Figure 1b). The distance of the structural unit in the x and y directions is P = 600 nm. The entire simulation was performed using the three-dimensional time-domain finite-difference method. The permittivity of Ag used in the calculation was obtained from the CRC database, and the permittivity of Si and $SiO_2$ was obtained from the Palik database, which was included in the simulation software (Lumerical FDTD Solutions). To simulate the structure accurately, we applied periodic boundary conditions in the x and y directions along with perfectly matched layer (PML) boundary conditions in the z-direction using a mesh accuracy of 3 nm in all three dimensions—x, y, z.

In the simulation, the absorption result of the absorber could be calculated by the equation $A(\lambda) = 1 - R(\lambda) - T(\lambda)$ [36], where $R(\lambda)$ and $T(\lambda)$ represent reflection and transmission, respectively. To understand the mechanism of the perfect absorber, we used the impedance matching principle. When the light source is fired vertically as shown in Figure 1, the reflection of the metamaterial absorber is [37]:

$$R = \left| \frac{Z - Z_0}{Z + Z_0} \right|^2 \tag{1}$$

where $Z_0$ is the free space impedance, Z is the effective impedance of the metamaterial absorber, and $Z_0$ and Z are denoted as follows:

$$Z_0 = \sqrt{\frac{\mu_0}{\varepsilon_0}} \tag{2}$$

$$Z = \sqrt{\frac{\mu}{\varepsilon}} \tag{3}$$

where $\mu_0$ and $\varepsilon_0$ denote the free-space permeability and permittivity, respectively, and $\mu$ and $\varepsilon$ denote the permeability and permittivity of the absorbing material, respectively [38–40]. Perfect absorption can be achieved when the reflectivity of the structure reduces to zero.

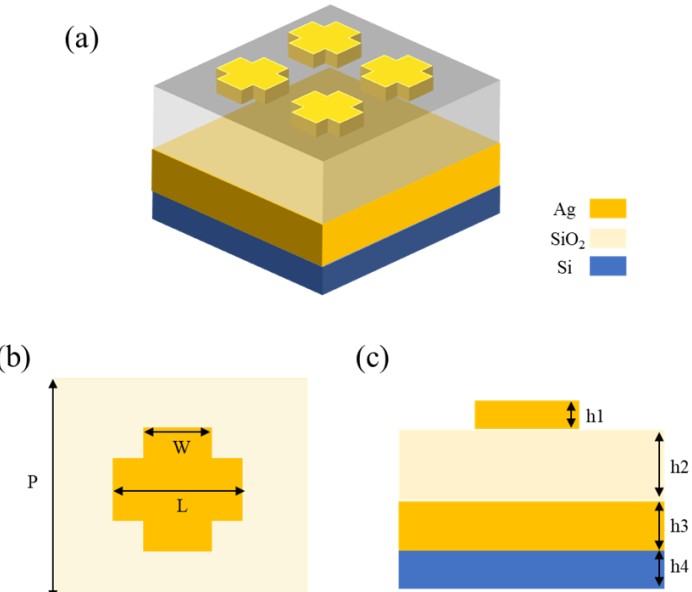

**Figure 1.** Two-band material absorber structure diagram: (**a**) Schematic diagram of the periodic array; (**b**) top view; (**c**) side view.

Based on Equations (1)–(3), when the surface impedance of a designed metamaterial structure matches that of air impedance ($Z = Z_0$), the reflection of the metamaterial structure becomes zero ($R = 0$). The coupling between the electric dipole and magnetic resonance or surface impedance in the metamaterial structure primarily depends on the thickness of the intermediate dielectric layer. Therefore, optimizing the layer's thickness allows us to achieve nearly 100% absorption of incident light. Through simulation calculations, we can preliminarily determine the geometric parameters that lead to perfect absorption. To achieve perfect absorption, it is crucial to ensure that the equivalent impedance of the structure equals 1. Since metals have negative permittivity at low frequencies, adjusting metal parameters becomes necessary to control their permittivity [40].

## 3. Results and Discussion

Figure 2 displays the absorption spectrum of the absorber, revealing two peaks at visible wavelengths $\lambda_1 = 654$ nm (peak 1) and $\lambda_2 = 781$ nm (peak 2), with absorbance levels reaching up to 99.83% and 99.64%, respectively. The absorption bandwidth of the absorption peaks is wide, which is good for information acquisition and processing.

To further elucidate the reasons behind the structure's broad and perfect absorption capabilities, the distribution of electric and magnetic fields is studied in Figure 3. The x-y plane shows electrical distributions in Figure 3a,b, while those in the x-z plane are shown in Figure 3c,d. Two resonance wavelength peaks at perpendicular incidence angles of 654 nm and 781 nm were observed on the structure. It can be seen that the electric field at the resonance wavelength is strongly concentrated in the cavity formed between the nano-metal cross-structure and the metal plate, which is an obvious gap plasmon resonance mode [41]. The nanostructures exhibit strong absorption at both wavelengths, resulting in an enhancement of the electric field by several orders of magnitude. For the gap plasmon resonance, the high-absorption property of the metal material and the high localized electric field distribution make the gap plasmon resonance mode have large inherent loss. As can be seen from the electric field distribution, most of the incident light energy is confined to the

metal structures, which results in the high absorption of the metallic material Ag [4]. As a result, the absorption peaks have a higher absorption intensity and a larger half-height width.

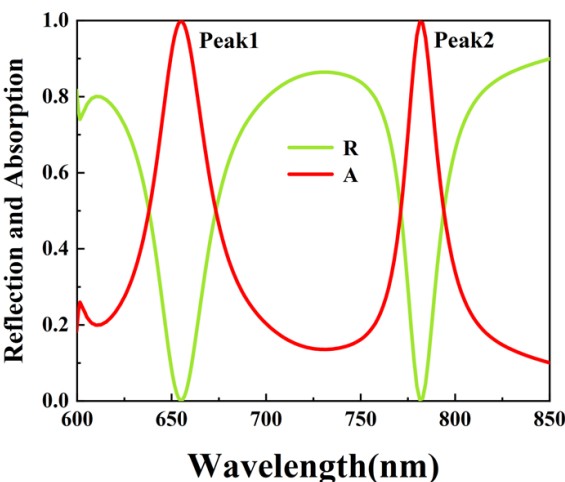

**Figure 2.** Reflection and absorption spectra.

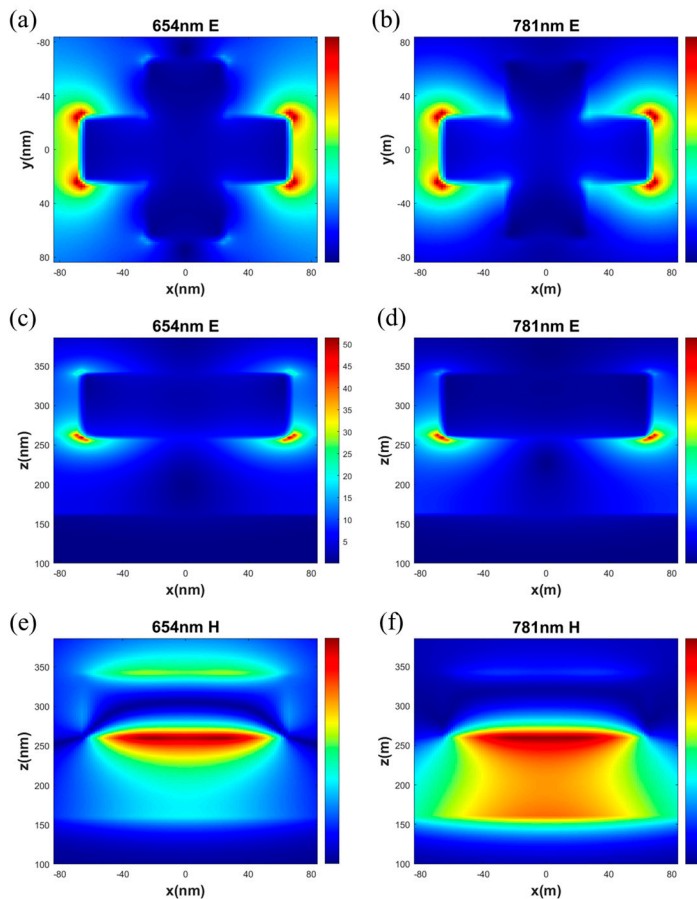

**Figure 3.** (**a**) The electric field diagram lies on xy plane at λ = 654 nm; (**b**) the electric field diagram lies on xy plane at λ = 781 nm; (**c**) the electric field diagram lies on xz plane at λ = 654 nm; (**d**) the electric field diagram lies on xz plane at λ = 781 nm; (**e**) the magnetic field diagram lies on xz plane at λ = 654 nm; (**f**) the magnetic field diagram lies on xz plane at λ = 781 nm.

However, the magnetic field distribution is significantly different. Specifically, at a wavelength of 654 nm (Figure 3e), we observed a strong confinement of the magnetic field between the top and bottom surfaces of Ag, confirming the excitation of localized surface

plasmon resonance [32]. The magnetic field at 781 nm is strongly confined to a very small space between the crossed silver metal top layer and the SiO₂ dielectric spacer layer. It is confirmed that the propagating surface plasmon (PSP) resonance was excited [41,42].

### 3.1. Modulation Analysis of the Period and Dielectric Thickness of the Absorber

The dielectric thickness of insulators is related to the absorption peak, as depicted in Figure 4a. When the thickness of the dielectric increases from 60 to 100 nm, the absorption peak position shifts from 746 to 781 nm, and the absorption also increases. By altering nanostructure period *P*, one can tune these absorption peaks accordingly, as demonstrated by Figure 4b. The *PSP* dispersion relation explains how changes in period *P* affect the *PSP* wavelength as follows [10,12]:

$$K_{psp} = \frac{2\pi}{\lambda_0} \times \sin\theta_{inc} + \frac{2\pi}{P} \times m \qquad (4)$$

where $K_{psp}$ is the *PSP* wave vector along the surface of Ag-SiO₂, and $\lambda_0$ denotes the wavelength of free space. $\theta_{inc}$ is the angle of incidence and m is an integer. When the period changes from 560 to 600 nm, there is an increase in absorption peak to the period, resulting in a shift in its position from 754 to 781 nm. The sensitivity towards period variations allows for the modulation of the absorption peak, making it suitable for applications across different frequency bands.

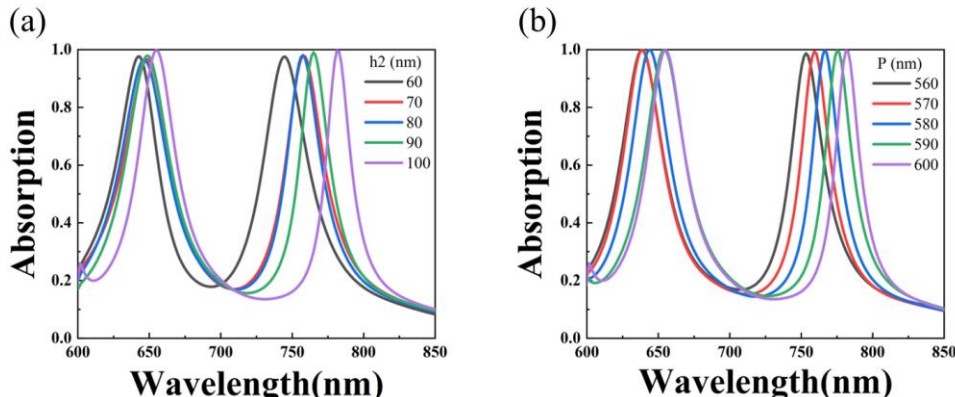

**Figure 4.** (**a**) Absorption spectra under different dielectric thicknesses; (**b**) absorption spectra of different structural periods.

### 3.2. Size Modulation Analysis of the Cross-Structure

Resonance was utilized to achieve a nearly unified absorption in the dual-band metamaterial absorber. The absorption frequency primarily depends on the structure's characteristic size, and by altering the size, it can modulate the position of the absorption peak. By changing the characteristic dimensions L and W of the structure, it is possible to change its absorption. Increasing L from 100 to 130 nm in increments of 10 nm results in a redshift of both absorption peaks from 625 to 654 nm and from 767 to 781 nm, respectively, while also gradually increasing the absorption intensity (Figure 5), which may be due to the increase in the effective index of Ag, which leads to the shift in the resonance wavelength to the long-band direction, and the increase in L, which leads to the increase in the Ag filling area relative to the whole structural unit. Therefore, with the increase in the plasmon pole on the surface of the Ag, the light absorption efficiency of the device also increases. When W, the characteristic size of the embedded structure, increases from 50 to 80 nm with an interval of 10 nm, it causes a blue shift in the absorption peak at 654 nm, moving it towards 647 nm. Additionally, there is a redshift in the absorption peak at long wavelengths as it changes from being centered at around 781 to approximately 787 nm; consequently, this leads to a gradual weakening of the overall absorption intensity, which is because with the increase in W, the pattern is too close to the adjacent structural units, mutual coupling occurs, and the excitation energy interferes with each other, resulting in a decrease in the absorption

rate. The individual changes do not affect each other, so independent adjustment can be achieved.

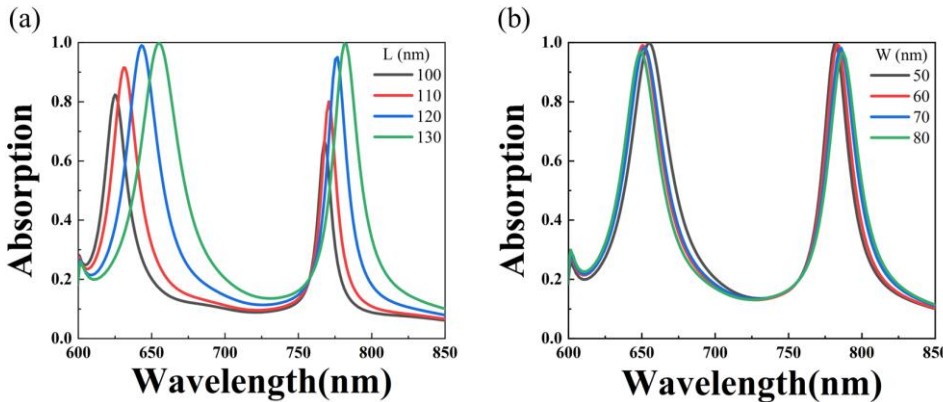

**Figure 5.** (**a**) Absorption spectra with diverse lengths for micro–nano structures; (**b**) absorption spectra with diverse widths for micro–nano structures.

### 3.3. Modulation Analysis of Refractive Index and Polarization Angle of the Absorber

Figure 6a shows how different refractive indices affect the absorber's absorbance curve when considering a fixed dielectric layer thickness of 100 nm. It can be observed that with an increase in the refractive index for a dielectric material, the absorption peak redshift and the absorption intensity of the short wavelengths increase slightly. The absorption peak of long wavelengths has an obvious redshift, and the absorption intensity is unchanged. Moreover, the long-wavelength peak changes linearly with the refractive index of the dielectric layer. The phenomenon arises due to the increased capacitance value resulting from a higher dielectric constant associated with the surrounding medium, resulting in the resonance shifting towards longer wavelengths.

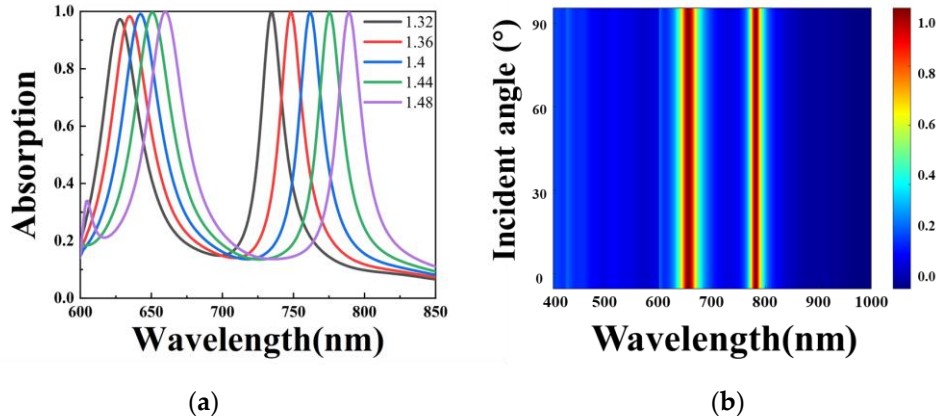

(**a**)  (**b**)

**Figure 6.** (**a**) Absorption spectra of different refractive indices of materials; (**b**) absorption spectra at different incident polarization angles.

When integrating our proposed metamaterial absorber into an imaging system together with a detector, it becomes crucially important to ensure the stability of the polarization angle. Yet, this aspect often gets overlooked. Figure 6b depicts how sensitive the absorber's absorption characteristics are to variations in the polarization angle. Under normal incidence conditions, the absorption spectrum results are shown in Figure 6b for when the polarization angle reaches between 0° and 90°. It is evident from the figure that the absorption spectrum remains unchanged as the polarization angle increases, indicating that the structure exhibits insensitivity to polarization. Moreover, the absorber can effectively operate in the visible-light range and find numerous applications, particularly in fields such as biology.

## 4. Conclusions

This study presents a two-band perfect metamaterial absorber that is insensitive to changes in polarization angle. The electromagnetic response characteristics of the absorber were studied theoretically and numerically. Approximate uniform absorption values of 99.83% and 99.64% were obtained at 654 nm and 781 nm, respectively. By studying the distribution of the electromagnetic field, it was found that the high absorption was caused by gap plasmon excitation. At 654 nm, absorption occurred due to the local surface plasmon resonance of the surface structure, while at 781 nm, absorption was generated by surface plasmon resonance. The thickness of the insulator and structural period primarily influence these absorption characteristics. Notably, this structure demonstrates excellent modulation properties. By adjusting the structure's characteristic size, the absorption conforms to the regular change. Alterations in the refractive index affect an increase in the dielectric constant within the structure, leading to a redshift caused by the increased capacitance value. Furthermore, it was observed that variations in the polarization angle did not impact the absorption spectrum, thus confirming its polarization-insensitivity property. Consequently, the proposed perfect absorber has many applications in the field of sensors in the visible-light region.

**Author Contributions:** Conceptualization, G.D. and Z.X.; formal analysis, Z.L., Z.X. and G.H.; funding acquisition, G.D.; writing, Z.X. and Z.L.; investigation, data curation, K.S. and Y.H. All authors have read and agreed to the published version of the manuscript.

**Funding:** This research was funded by the 2021 Sichuan Provincial Philosophy and Social Science Key Research Base Research Project (CJF21012) and Chengdu Normal University horizontal project (2024HX07, 2022HX60).

**Institutional Review Board Statement:** Not applicable.

**Informed Consent Statement:** Not applicable.

**Data Availability Statement:** The data provided in this study may be obtained from the corresponding author upon reasonable request.

**Conflicts of Interest:** The authors declare no conflicts of interest.

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
