# Peer review of "Polarization-Angle-Insensitive Dual-Band Perfect Metamaterial Absorbers in the Visible Region: A Theoretical Study"

_coatings, doi:10.3390/coatings14020236_

Round 1

Reviewer 1 Report

Comments and Suggestions for Authors

This manuscript reports the Polarization angle insensitive dual-band perfect metamaterial absorbers in the visible region. The authors studied a polarization Angle insensitive dual-band perfect metamaterial absorber with absorption peaks of 654 nm and 781 nm, respectively. By adjusting the structure parameters, dielectric thickness, and refractive index, the obtained absorber has high scalability in the visible wavelength region. To further understand the performance of the cross-structure absorber, the analysis of its electric and magnetic field distribution shows that it produces two resonance modes leading to different absorption properties. In addition, the position and intensity of the absorption peaks were found to be unchanged with increasing incident polarization angle, indicating that the absorber is insensitive to the polarization of the incident light. The absorber has great flexibility and has a good application prospect in sensing and detection. In my opinion the manuscript is best for Coatings, after the authors have addressed the following major comments.

11.      All figures must be improved and replete again.

2.      What is the primary focus of the paper's investigation?

3.      Why are metamaterial absorbers significant in the field of photonics?

4.      Can you describe the proposed dual-band perfect metamaterial absorber's absorption peaks and their corresponding wavelengths?

5.      How is the scalability of the absorber achieved in the visible wavelength region?

6.      What parameters are adjusted to achieve the desired properties of the dual-band perfect metamaterial absorber?

7.      How does the electric and magnetic field distribution analysis contribute to understanding the performance of the cross-structure absorber?

8.      What are the two resonance modes produced by the cross-structure absorber, and how do they influence absorption properties?

9.      What role do structure parameters, dielectric thickness, and refractive index play in the absorber's performance?

10.  How does the obtained absorber exhibit high scalability in the visible wavelength region?

11.  Why is the analysis of the absorption peaks' position and intensity crucial in understanding the absorber's behavior?

12.  What does it mean for the absorber to be insensitive to the polarization of incident light?

13.  How does the proposed absorber demonstrate flexibility, and what are its potential applications mentioned in the abstract?

14.  Can you elaborate on the significance of the absorber's application prospect in sensing and detection?

15.  In what ways does the proposed dual-band perfect metamaterial absorber contribute to advancements in photonics research?

  1.  

Reviewer 2 Report

Comments and Suggestions for Authors

Authors present an interesting study on a metamaterial based in a heterostructure made from silver, silicon and SiO2. The manuscript is well written and organized, however, several issues should be addressed before acceptance.

Please find all my comments in the marked attached pdf file.

Comments on the Quality of English Language

English is good, however there are several mistakes that should be corrected

Reviewer 3 Report

Comments and Suggestions for Authors

Review

 Polarization angle insensitive dual-band perfect metamaterial absorbers in the visible region

General: The paper presents interesting results but does not show at all how to get there. In particular:

1.      Eq. 2 , 3 and 4 are circles, the parameter Z is explained by itself.

2.      I guess that figure 2 is used by calculation from eq. 2-4, but the matrix operators must be defined diferently

3.      Fig. 3: there is no explanation at all how these figures were obtained

4.      Eq. 7: Period p, PSP, Kpsp, are not explained in detail and are used diferently P instead of p, Kpsp instead of Kpçp in the text. In section 2 the period P is called distance.

5.      Fig.4 and 5: From the text it is understood that the geometry (width, length, thickness ) is influencing the absorption. Which equation was used for those calculations?

6.      Figure 6: index of refraction variation of which material SiO2 layer????

7.      Figure 6: How does the polarization abgle enters into the calculations

The paper needs a major revision showing the path from material data (geometry, complex refractive índices of all materials) to the results of the calculation.

Comments on the Quality of English Language

The english language is usage is understanble. Some phrases could be shortened.

Reviewer 4 Report

Comments and Suggestions for Authors

This paper deals with simulations of the optical response of "metamaterial absorber" structures, a topic that has been the subject of numerous recent publications, some of them very recent and not cited by the authors (https://doi.org/10.3390/coatings12050687, https://doi.org/10.1364/OE.437435, https://doi.org/10.1016/j.ijleo.2023.170915 ).

The first criticism of this type of work is the absence of any confrontation with experimental measurements to corroborate the results of the calculations and confirm their applicability.

The second criticism is the absence of any indication of the dielectric functions used in the calculations for the materials (Ag, SiO2, Si) that make up these structures. This is an important issue, since at nanometric thicknesses the optical properties of thin films can be different from those of thick materials.

The polarization insensitivity at normal incidence can be explained by the z-axis symmetry of the structure. The authors have varied various parameters (dielectric thickness and periodicity of the structure) on the positions of the two absorption bands. However, no explanation is given for the existence of these two absorption bands and their relationship to the choice of materials making up the structure.

Some typing errors and imperfections to be corrected:

- From a practical point of view, a Si substrate to support and protect the entire device must have a thickness h4 of a few hundred micrometers (the thickness of a Si wafer) and not 200 nm.

- The fractions in equation 4 need to be written in a more presentable way.

Round 2

Reviewer 1 Report

Comments and Suggestions for Authors

accepted 

Author Response

Dear   reviewer:

thank you for your comments!

Reviewer 3 Report

Comments and Suggestions for Authors

Equations 3,4,9 are still circular equations. (The answer by the author confrms that they are circular, but no changes) The paper pretends to calculate reflection, absorption for multilayered structured coatings, which is an interesting topic, interesting results are shown, but, in my opinion, should not be published without supporting equations. These supporting equations are still missing. The explanation by the authors are general.

Reviewer 4 Report

Comments and Suggestions for Authors

The authors have more or less responded to my comments. However, the revised manuscript with the changes and deletions crossed out is still difficult to read.

In order to make it clear to the reader that this work is not experimental but rather a numerical simulation, I would like the authors to indicate this in both the title of the paper and the abstract.

For the title, I suggest "Polarization angle insensitive dual-band perfect metamaterial absorbers in the visible region: a theoretical study".

For the abstract: "In this paper, we propose a simulation study of a polarization angle insensitive dual-band perfect metamaterial absorber with absorption peaks at 654 and 781 nm, respectively".

Round 3

Reviewer 3 Report

Comments and Suggestions for Authors

Refrence 39 is not complete

Reference 40, from the same University, provides much better mathematical description of similar experimental set-up

There are no substancial changes to the manuscript, as suggested. From my perspective, the pblication is incomplete because it cannot be reproduced by the reader. Apparently, see reference 41, this is aceptable. It is a pitty that the major equations are not included, because these would make the publication much richer. 
